# Complete and Incomplete Pentalogy of Cantrell

**DOI:** 10.3390/children6100109

**Published:** 2019-10-06

**Authors:** Ranjit I. Kylat

**Affiliations:** Department of Pediatrics, University of Arizona, College of Medicine, PO BOX 245073, 1501 N Campbell Avenue, Tucson, AZ 85724, USA; rkylat@gmail.com; Tel.: +1-520-626-6627; Fax: +1-520-626-5009

**Keywords:** pentalogy of Cantrell, omphalocele, ectopia cordis, congenital heart disease

## Abstract

Pentalogy of Cantrell (PC) is a malformation characterized by defects in the ventral abdominal wall, lower sternum, diaphragmatic pericardium, anterior diaphragm associated with omphalocele, thoraco-abdominal ectopia cordis, diaphragmatic hernia, and intracardiac abnormalities. PC is stratified as complete or incomplete and we present both the complete and incomplete forms.

## 1. Introduction

Pentalogy of Cantrell (PC) is a malformation characterized by defects in the ventral abdominal wall, lower sternum, diaphragmatic pericardium and anterior diaphragm associated with omphalocele, thoraco-abdominal ectopia cordis, diaphragmatic hernia, and intracardiac abnormalities, all of which were present below in Case 1 (clinical and autopsy findings) [1]. Here we describe both the complete and incomplete forms of PC. 

## 2. Description

### 2.1. Case 1

A 39-week gestation infant was born to a 20-year-old primigravida whose pregnancy was complicated by pre-gestational diabetes mellitus and late prenatal care starting at 31 weeks, when the fetus was diagnosed with PC. Prenatal ultrasound showed a large omphalocele with herniation of liver, anterior diaphragmatic hernia, absence of lower sternum, ectopia cordis, and a ventricular septal defect (VSD). Maternal laboratory tests were all normal and amniocentesis revealed 46 XX on karyotype. She was on prenatal vitamins, insulin, and metformin, which was discontinued at the time of conception. The mother had no history of teratogen exposure, smoking, alcohol or illicit drugs. Apart from a maternal aunt with Holt-Oram syndrome there was no other relevant family history. There was no history of consanguinity and the father was 22 years old. The infant was born via vaginal delivery with APGAR of five and eight at one and five minutes and required mechanical ventilation for respiratory distress. The infant’s birth weight was 2790 g (25th percentile), length was 46 cm (25th percentile), head circumference was 33.5 cm (50th to 75th percentile). Examination revealed a term infant with a small abdomen and thorax with the internal organs herniated into a large omphalocele with bowel and liver visible through the sac, a pulsatile ectopia cordis with visible pericardium and heart outside the thorax. Chest radiographs showed absent middle and lower sternum, and bowel in the anterior part of the chest (Figure 1 and Figure 2). Echocardiography showed an anomalous right coronary artery from the left sinus, large peri-membranous VSD, partially closed with tricuspid septal leaflet tissue, patent ductus arteriosus (PDA), and mild sub-valvular pulmonic stenosis. In addition, systemic right ventricular pressure, as measured by the tricuspid regurgitation jet, suggested pulmonary arterial hypertension. She was weaned off mechanical ventilation and enteral feeds were started at three days, but she had intolerance with emesis and was again attempted after a few days. Investigations carried out to determine the extent of the prenatal findings determined it to be consistent with Pentalogy of Cantrell, including sternal cleft, anterior diaphragmatic hernia, ectopic cordis, ventricular septal defect, and a large omphalocele. Her microarray was normal. After ten days she had an operative procedure, where an attempt to separate her bowel from her heart was made. The large omphalocele was enclosed with a Gore-Tex synthetic patch (W. L. Gore Inc., Flagstaff, AZ, USA) along with closure of the sternal defect, mediastinal defects, and anterior abdominal wall defects. She had broad-spectrum antibiotics peri-operatively and she remained on mechanical ventilation after the operative procedure. A few days later she developed infectious complications around the Gore-Tex patch and the wound culture grew *Stenotrophomonas* and methicillin-sensitive *Staphylococcus aureus*. She needed escalation of her respiratory support and was hemodynamically very unstable needing to be resuscitated twice for severe decompensation. At 19 days, the parents elected to withdraw critical care support with demise soon afterwards.

Autopsy revealed that the organs of the thorax and abdomen were not in their normal anatomic relationships with the heart protruding beneath the rib cage and sternum, with the liver and intestines protruding from the abdominal cavity, but the large abdominal and thoracic wall defect were covered by surgical Gore-Tex. The underlying liver and abdominal organs were severely adhered with abscesses below and above the liver with culture confirming the growth of *Staphylococcus aureus*, *Stenotrophomonas* (*Xanthomonas*) *maltophilia*, and *Enterococcus fecalis*. An acute organizing pericarditis and serous fluid in both pleural cavities were identified. The heart was grossly elongated. There were large atrial and ventricular septal defects, and a large patent ductus arteriosus was identified. The liver showed centrilobular congestion along with extra-medullary hematopoiesis and multiple abscesses. The findings of PC after Gore-Tex closure of sternal and abdominal wall defect, ectopia cordis and omphalocele along with absence of distal sternum, anterior diaphragm, and diaphragmatic pericardium were confirmed. 

### 2.2. Case 2

A male infant born at 38 weeks was transferred soon after birth for multiple anomalies. His mother, a 22-year-old, had prenatal care and was not aware of any abnormalities prior to delivery. There was no relevant family history and no exposure to teratogens or a history of consanguinity. The weight at birth was 2400 g and anomalies were noted necessitating the transfer. Physical examination revealed ectopia cordis and umbilical hernia with no other external anomalies or dysmorphism (Figure 3). The chest radiograph showed ossification of the manubrium and the upper sternal segment, however with deficient ossification of the lower sternum (Figure 4). Echocardiography showed a small secundum atrial septal defect and a left superior vena cava draining to a dilated coronary sinus. Ultrasound (US) of the head was normal and US of abdomen revealed small bowel contained within umbilical hernia without evidence of obstruction or other anomalies. (Figure 5) The patient had a normal routine metabolic screen and the microarray was normal. The patient remained stable in room air and tolerated feeds. The patient was discharged home and was scheduled to have outpatient surgery at a later age.

Procedures performed in studies involving human participants were in accordance with the ethical standards of the institutional and/or national research committee and with the Helsinki Declaration. The authors retain an informed consent from parents for the autopsy performance, and for publication and the study was approved by the institutional ethics research committee.

## 3. Discussion

Pentalogy of Cantrell (PC) is an extremely rare association of congenital anomalies originally described in 1958 [1]. It has also been called Cantrell syndrome or deformity, thoraco-abdominal syndrome, and X-linked midline defect. The incidence of occurrence is estimated in the literature to be from 1 in 65,000 to 1 in 200,000 live births [2,3,4].

It is more commonly found in males and has multifactorial inheritance. The embryologic basis is thought to be a defective formation, differentiation and migration of mesoderm in the early embryonic period. There is a failure in the folding mechanism of the lateral body folds and cranio-caudal folds that converts the flat trilaminar germ disc into a tubular structure starting at about 5 weeks gestation and which normally should converge and contract at the umbilical ring, thereby closing the ventral abdominal wall [5,6]. This could be due to aberrant development and migration of muscular components of the abdominal wall and the defect is cephalad to the umbilicus [5,6]. PC is classified as a definite diagnosis or as complete or incomplete based on the number of malformations present [3]. Class 1 is a definite diagnosis or complete syndrome when all five defects are present; Class 2 is a probable diagnosis, when four of the defects are present, including intracardiac and ventral wall abnormalities; and Class 3 is an incomplete expression, when some of the defects are present, and generally should include the sternal abnormality [7].

A wide range of intracardiac anomalies have been associated with PC including septal defects, tetralogy/pentalogy of Fallot, total anomalous pulmonary venous drainage, and left ventricular diverticulum [7,8,9,10,11]. Holt-Oram syndrome, which was seen in the family of Case 1, is associated with cardiac anomalies but has never been reported with PC. 

A variety of mutations including deletions or duplications of the PORCN, thoraco-abdominal syndrome (TAS), ALDH1A2, and teneurin-4 (TENM4) genes have been identified [12,13,14,15]. It can also be associated with rare anomalies, such as craniorachischisis, pulmonary extrophy, and Goltz-Gorlin syndrome [16,17,18,19].

## 4. Conclusions

PC is rare association of anomalies and can be challenging to manage. Prenatal diagnosis and careful assessment is essential for appropriate postnatal management [20,21]. A multidisciplinary team including pediatric surgery, cardiac and thoracic surgery is essential for optimal management [22,23,24,25,26]. It would be prudent to delay surgical interventions to a later date if hemodynamically stable and if enteral feeding can be fully established. A staged surgical approach is better, if feasible, and in cases where synthetic patch closure is needed, as the risk of infection including abscesses are extremely high.

## Figures and Tables

**Figure 1 children-06-00109-f001:**
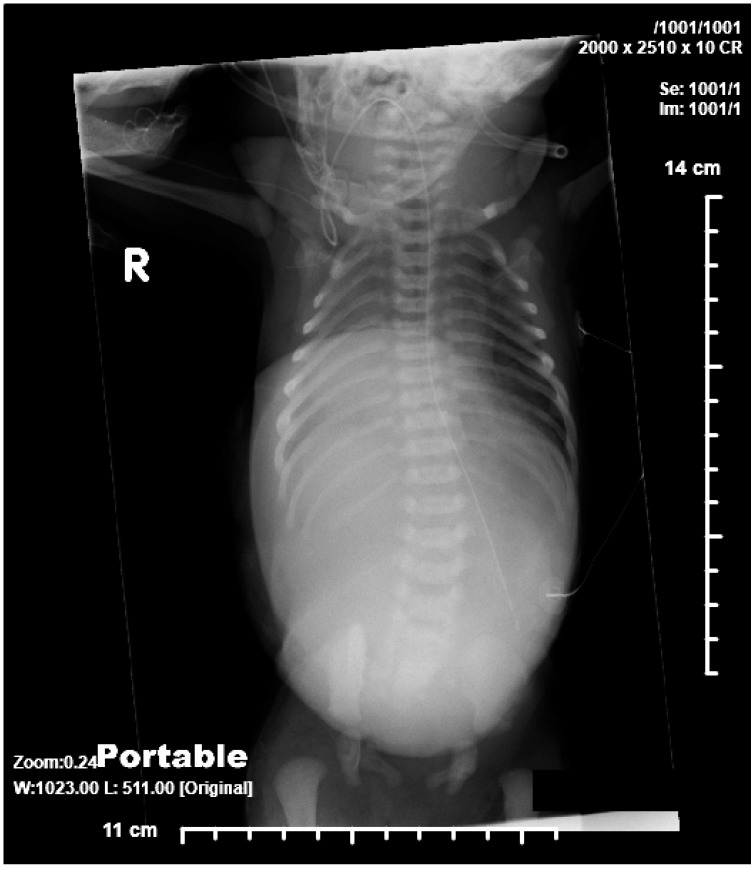
AP radiograph showing absence of lower part of sternum, large omphalocele, and suggestion of pulmonary hypoplasia.

**Figure 2 children-06-00109-f002:**
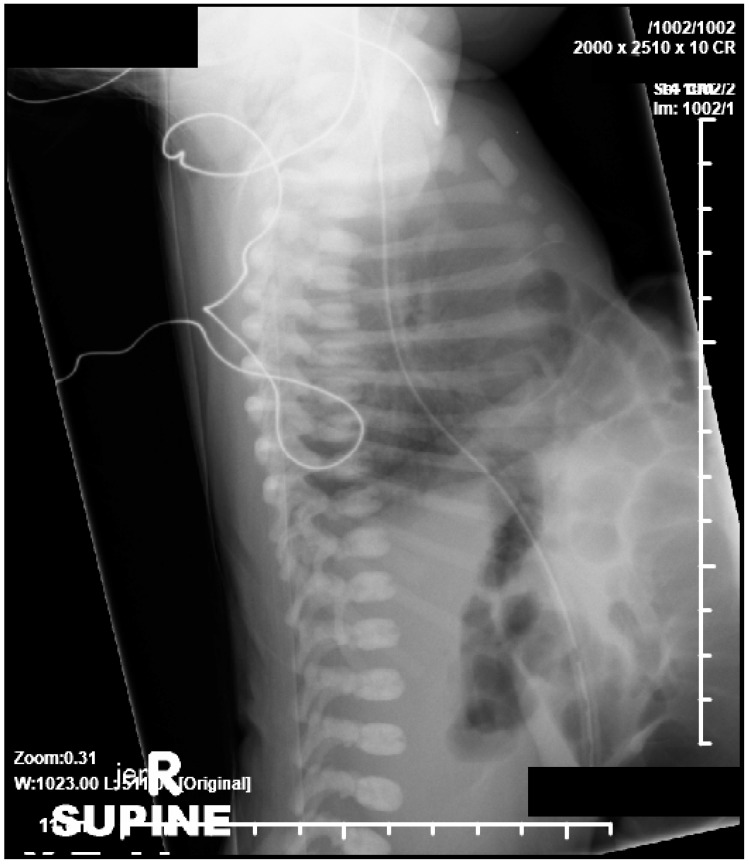
Lateral radiograph showing absence of lower part of sternum, large omphalocele, anterior diaphragmatic herniation, and suggestion of pulmonary hypoplasia.

**Figure 3 children-06-00109-f003:**
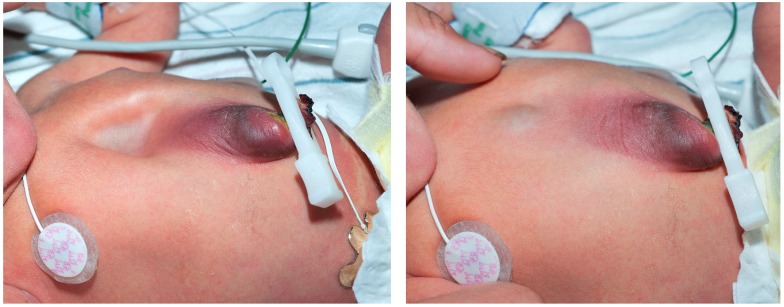
Photograph during inspiration and exhalation revealing ectopic cordis and minor omphalocele.

**Figure 4 children-06-00109-f004:**
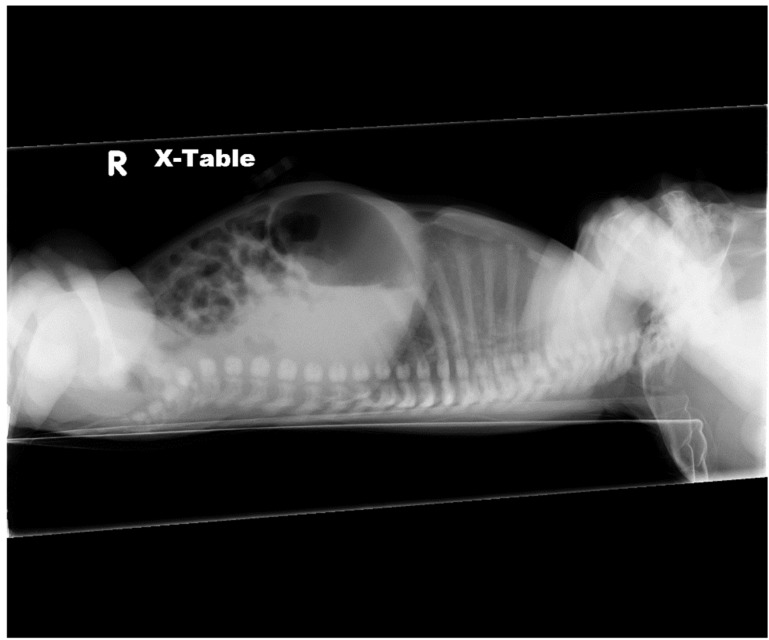
Lateral radiograph showing agenesis of lower third of sternum.

**Figure 5 children-06-00109-f005:**
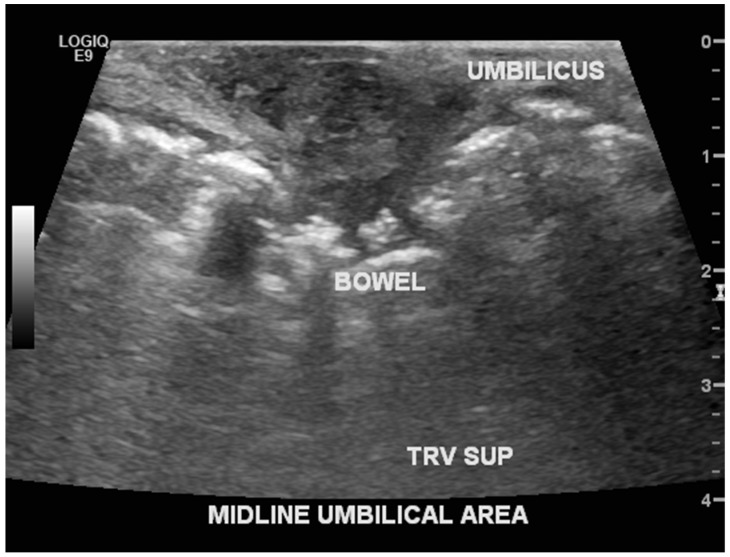
Abdominal ultrasound revealing intestine in the herniated part of umbilicus-omphalocele minor.

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
