# Peer review of "Complete and Incomplete Pentalogy of Cantrell"

_children, 2019, doi:10.3390/children6100109_

Round 1

Reviewer 1 Report

The author presents a complete report of 2 cases of Pentalogy of Cantrell.  Such cases are rare but reported in the literature.  The author's presentation of both complete and incomplete Pentalogy provides the reader the range of presentation and outcome.

I can provide a number of small suggestions:

Line 29: I think it is worth noting for the readers that Holt-Oram syndrome is associated with cardiac defects but has no reported relation to PC. Line 38: Please describe "ectopic" coronary artery.  This description of coronary anatomy is not usual.  I would recommend a more-descriptive "anomalous left coronary artery from the right sinus," "single coronary artery origin," or the like. Line 40: The mention of systemic pulmonary artery pressures should be qualified if it was thought to be of significance.  Was this pressure thought to be due to usual neonatal transition (eg. resolved over the days of transition) or was there a suggestion of pulmonary arterial hypertension.  Additionally, direct estimate of pulmonary ARTERY pressure by echocardiography is difficult.  I would presume your echocardiogram estimated right ventricular pressure by tricuspid valve regurgitation Doppler.  I would recommend altering the wording to "systemic right ventricular pressure." Line 46/49: Gore-Tex is a product name.  Manufacturer details should be included.  No need to place Gore-Tex in quotes.

Author Response

I am extremely obliged to you for taking the time to review, edit and provide impartial and constructive criticisms and suggestions. I have tried to incorporate all the suggestions now, but if there are any omissions I would gladly do that at the next phase.

Line 29: I think it is worth noting for the readers that Holt-Oram syndrome is associated with cardiac defects but has no reported relation to PC.

Thanks very much. I have added this information in the discussion.

Line 38: Please describe "ectopic" coronary artery. This description of coronary anatomy is not usual.  I would recommend a more-descriptive "anomalous left coronary artery from the right sinus," "single coronary artery origin," or the like.

I have verified the original chart and confirm that the correct version was as you described and I have changed that in the description of the case.

Line 40: The mention of systemic pulmonary artery pressures should be qualified if it was thought to be of significance. Was this pressure thought to be due to usual neonatal transition (eg. resolved over the days of transition) or was there a suggestion of pulmonary arterial hypertension.  Additionally, direct estimate of pulmonary ARTERY pressure by echocardiography is difficult.  I would presume your echocardiogram estimated right ventricular pressure by tricuspid valve regurgitation Doppler.  I would recommend altering the wording to "systemic right ventricular pressure."

Thanks very much. Verified and corrected.

Line 46/49: Gore-Tex is a product name. Manufacturer details should be included.  No need to place Gore-Tex in quotes.

Thanks once again. The edits have been made in this current version.

Reviewer 2 Report

I have included comments on my edited draft of the manuscript.

There are a few sentences that seem to not have importance or I do not understand their importance. There are a few grammar issues that I also address.

I am wondering about why this is felt to be important and novel? I think that needs to be stated. PC is very rare and can be very difficult to manage. What about these two cases being presented in the literature helps the next set of practitioners in their care/management of this problem?

Author Response

I have included comments on my edited draft of the manuscript.

I am extremely obliged to you for taking the time to review, edit and provide impartial and constructive criticisms and suggestions. I have tried to incorporate all the suggestions now, but if there are any omissions I would gladly do that at the next phase.

There are a few sentences that seem to not have importance or I do not understand their importance. There are a few grammar issues that I also address.

Thanks very much.

I am wondering about why this is felt to be important and novel? I think that needs to be stated. PC is very rare and can be very difficult to manage.

Thanks very much. I have included the statement in the conclusions.

What about these two cases being presented in the literature helps the next set of practitioners in their care/management of this problem?

I have tried to address this but have not been fully able to- added statement regarding the difficulty in management and the need for a multidisciplinary staged management. There is also an addition about delaying surgical interventions and avoiding synthetic patches.